# Pathological Heart Rate Regulation in Apparently Healthy Individuals

**DOI:** 10.3390/e25071023

**Published:** 2023-07-05

**Authors:** Ludmila Sidorenko, Irina Sidorenko, Andrej Gapelyuk, Niels Wessel

**Affiliations:** 1Department of Molecular Biology and Human Genetics, State University of Medicine and Pharmacy, “Nicolae Testemitanu”, Stefan cel Mare Str. 165, MD-2004 Chisinau, Moldova; ludmila.sidorenco@usmf.md; 2Medical Center “Gesundheit”, Mihai Kogalniceanu Str. 45/2, MD-2009 Chisinau, Moldova; 3Cardiovascular Physics, Humboldt-Universität zu Berlin, D-10099 Berlin, Germany; 4MSB Medical School Berlin GmbH, D-14197 Berlin, Germany

**Keywords:** heart rate variability, cardiorhythmogram, pathological appearing heart rhythm regulation, non-linear dynamics, risk prediction

## Abstract

Cardiovascular diseases are the leading cause of morbidity and mortality in adults worldwide. There is one common pathophysiological aspect present in all cardiovascular diseases—dysfunctional heart rhythm regulation. Taking this aspect into consideration for cardiovascular risk predictions opens important research perspectives, allowing for the development of preventive treatment techniques. The aim of this study was to find out whether certain pathologically appearing signs in the heart rate variability (HRV) of an apparently healthy person, even with high HRV, can be defined as biomarkers for a disturbed cardiac regulation and whether this can be treated preventively by a drug-free method. This multi-phase study included 218 healthy subjects of either sex, who consecutively visited the physician at Gesundheit clinic because of arterial hypertension, depression, headache, psycho-emotional stress, extreme weakness, disturbed night sleep, heart palpitations, or chest pain. In study phase A, baseline measurement to identify individuals with cardiovascular risks was done. Therefore, standard HRV, as well as the new cardiorhythmogram (CRG) method, were applied to all subjects. The new CRG analysis used here is based on the recently introduced LF drops and HF counter-regulation. Regarding the mechanisms of why these appear in a steady-state cardiorhythmmogram, they represent non-linear event-based dynamical HRV biomarkers. The next phase of the study, phase B, tested whether the pathologically appearing signs identified via CRG in phase A could be clinically influenced by drug-free treatment. In order to validate the new CRG method, it was supported by non-linear HRV analysis in both phase A and in phase B. Out of 218 subjects, the pathologically appearing signs could be detected in 130 cases (60%), *p* < 0.01, by the new CRG method, and by the standard HRV analysis in 40 cases (18%), *p* < 0.05. Thus, the CRG method was able to detect 42% more cases with pathologically appearing cardiac regulation. In addition, the comparative CRG analysis before and after treatment showed that the pathologically appearing signs could be clinically influenced without the use of medication. After treatment, the risk group decreased eight-fold—from 130 people to 16 (*p* < 0.01). Therefore, progression of the detected pathological signs to structural cardiac pathology or arrhythmia could be prevented in most of the cases. However, in the remaining risk group of 16 apparently healthy subjects, 8 people died due to all-cause mortality. In contrast, no other subject in this study has died so far. The non-linear parameter which is able to quantify the changes in CRGs before versus after treatment is FWRENYI4 (symbolic dynamic feature); it decreased from 2.85 to 2.53 (*p* < 0.001). In summary, signs of pathological cardiac regulation can be identified by the CRG analysis of apparently healthy subjects in the early stages of development of cardiac pathology. Thus, our method offers a sensitive biomarker for cardiovascular risks. The latter can be influenced by non-drug treatments (acupuncture) to stop the progression into structural cardiac pathologies or arrhythmias in most but not all of the patients. Therefore, this could be a real and easy-to-use supplemental method, contributing to primary prevention in cardiology.

## 1. Introduction

Cardiovascular diseases are the leading cause of morbidity and mortality in adults worldwide [1], although ongoing development is enabling more and more modern treatment techniques. Therefore, more resources should be invested in the prediction of structural heart diseases and arrhythmias, as well as in the research of prophylactic measurements [1,2,3]. A possible solution for prediction is offered by physiology. From a physiological point of view, before a structural heart disease or arrhythmia is manifested, these are anticipated by a latent period of disturbances of the heart regulation [4,5,6]. Usually, a structural heart disease is manifested when all of the compensatory mechanisms of regulation of the heart at the medullary and the central levels are broken down [7,8]. The person then enters a dangerous state, where the heart regulation migrates in a compensatory sense from the medullary to the predominant central level of regulation [4,9,10]. It is obvious that such a migration of heart regulation should be recognized as early as possible in order to prevent further progression into a structural heart disease or arrhythmia [3,8,11,12].

One of the well-known methods nowadays for assessing the state of the neural regulatory systems of the heart is the heart rate variability (HRV) [6,10,13,14,15]. Applying this method, when judging the reliability of the transition from the medullary level of regulation to the central one, shows a lot of limitations [8,13,16,17]. One of the important limitations is the appearance of non-steady-state events in a steady-state cardiorhytmogram (CRG—time series of beat-to beat-intervals; also called tachogram) [18,19]. This is because nonstationary HRV, analyzed when using standard linear methods, does not always show valid and credible results [8,13,16,18,19]. In the present study, a recently introduced CRG approach is offered by using dynamical physiological methods. Another essential problem when applying classical linear methods of HRV for the appreciation of the physiological state of the heart’s regulation, e.g., for prognosis construction, is that it is mainly based on the conclusion: whether the HRV is high or low [16]. In this context, it is important to keep in mind that the HRV decreases significantly when all of the physiological compensatory mechanisms are broken down [6,11,12,20]. From a clinical point of view, this means that, under such conditions, the patient starts showing corresponding morbidity manifestations [11,21,22,23]. This is why a classical linear analysis cannot be applied for primary prophylaxis or a reliable prediction of cardiovascular risks in healthy individuals [3,12,24].

In this study, we searched for methods which would enable a reliable detection of biomarkers for pathological heart regulation in still-healthy persons even with a high HRV. Additionally, to support the realization of this aim, the CRGs were analyzed by non-linear methods. Non-linear methods [3,15,17,25,26,27,28] are recently introduced methods to measure HRV and are not affected by non-stationarity, as linear indexes are. They include a.o. power law exponent, approximate entropy, and detrended fluctuation analysis. These methods study all complex interactions of hemodynamic, electrophysiological, and humoral variables as well as the autonomic and central nervous regulations.

Therefore, primarily, the aim of this study was to find out whether certain signs which show a pathological cardiac regulation, detected in a CRG of an apparently healthy person, even with high HRV, can be influenced clinically. Secondly, the study aimed to find out which non-linear parameter can characterize the pathophysiological signs identified in CRG, supporting the automatic CRG assessment for risk prediction.

## 2. Material and Methods

In this study, 218 healthy persons were included, who consecutively visited the physician at medical center “Gesundheit” because of arterial hypertension, depression, headache, psycho-emotional stress, feeling of extreme weakness, disturbed night sleep, feeling of heart palpitations, or chest pain. However, physical examinations, i.e., ECG, stress test, and echocardiography, did not reveal any structural cardiovascular diseases. The general duration of the study was three years. Both genders were included, female 65% (N = 142). The average age was 42.8 ± 12.0 years, the BMI was 25.11 ± 4.9.

The inclusion criteria were echocardiographic healthy persons, i.e., persons without structural heart diseases in sinus rhythm during the measurement. The exclusion criteria were: age under 18 years, arrhythmias, structural heart diseases, pregnancy, lactation period, menopause, influenza, thyrotoxicosis, medication which could influence the HRV, Parkinson disease, and any kind of state where tremor of extremities was observed.

From all subjects included, the HRV was analyzed by linear as well as non-linear methods, including the pathophysiologic CRG features. Therefore, the biosignal recording was done in the form of a 5-min steady-state ECG. Hence, the ECG was recorded using the specialized hardware Polyspectrum. All standard rules and procedures during the measurement were applied [8,13,16]. During the preparation of the persons for the measurement, all standard rules and procedures were also respected, most prominently the reaching of steady state before starting the measurement itself [13,16,18]. The physiologic method of cardiorhythmogram analysis is based on features’ analysis and is dependent on the person who analyzes the features. In order to approve this process and to create an automatized basis for the prediction of cardiovascular risks by analyzing a CRG, it was also analyzed by standard linear and existing non-linear methods of HRV analysis. The parameters applied for the linear methods of HRV analysis were meanNN, sdNN, sdaNN1, rmssd, pNN50, cvNN, VLF, LF, and HF [13,16,18,19]. To the non-linear HRV parameters belong Shannon, noNNtime, FORBWORD, FWSHANNON, FWRENYI4, WSDVAR, WPSUM02, and WPSUM13 [17,27,28,29]. FORBWORD stands for the symbolic dynamics parameter number of “forbidden words” in the distribution of words with length three: that is, the number of words which never or only seldomly occur. A high number of forbidden words stands for a rather regular behavior in the time series. If the time series is highly complex then, when using Shannon, only a few forbidden words will be found. FWSHANNON and FWRENYI4 denote the Shannon resp. Renyi entropy (of order 4) is calculated from the distribution of words in the same symbolic dynamics and both are classical complexity measures in time series. Higher entropy values refer to higher complexity in the corresponding tachograms and lower values to lower ones. WSDVAR is defined as the variability of different types of words occurring in the given symbolic dynamics. WPSUM02 and WPSUM13 quantify the percentage of words with low variability (02) as well as with high variability (13). The physiologic CRG analysis takes certain features of the waves of the heart rate time series—the low frequency (LF) drops and the high frequency (HF) exhibits counter-regulation. They can be quantified statistically by FWRENYI4 because the Renyi entropy of order 4 estimates the global complexity of the tachograms—rarely occurring words have no weight.

The study was performed in several stages. In stage A, all individuals were subject to the initial HRV measurement, followed by HRV analysis. This was performed step by step: firstly, the HRV was analyzed using standard linear methods; secondly, a physiological CRG analysis and non-linear analysis was performed. The aim of stage A was to divide all subjects into a group of healthy individuals and a risk group. The division was done based on the standard HRV as well as on the physiologic CRG method. According to the described HRV application standards [8,16], based on the linear HRV, the individuals were divided into the risk group if the HRV was low, and into the healthy group if the HRV was high [8,16,18]. HRV was defined as high if pNN50 > 0.14; it was considered as low if pNN50 ≤ 0.14. The physiologic CRG analysis allocated the persons with identified pathological signs in the cardiorhythmogram to the risk group. In the case of no pathological signs in the cardiorhythmogram, the individuals were classified into the healthy group.

In the next stage, stage B, it was proved whether the pathological signs identified in stage A can be influenced clinically by a drug-free treatment. It was important to find out whether the signs which were recognized as abnormal in the cardiorhythmograms, indicating a pathological state of the heart’s regulation, could be treated before they would progress into structural heart disease or arrhythmia. Acupuncture was chosen as a physiological drug-free treatment. Therefore, all subjects from the risk group were treated.

In the next stage, stage C, a comparative measurement of HRV by linear, non-linear, and physiological CRG analysis, immediately following the end of the acupuncture treatment course, was done. The chosen treatment method corresponded to some important criteria: it should be a non-pharmacological treatment, a non-psychological and non-psychiatric treatment, and a treatment which corresponds to the demand of reproducibility and, henceforth, could be further applied in order to prevent the progression of the pathological signs. One type of possible treatment, corresponding to these criteria, is acupuncture [30,31]. In the study conducted by Sroka K., it is reported that acupuncture can be applied as a reliable drug-free alternative for restoring the physiological functional state of the parasympathical component of the VNS, being reflected in the stabilization of the heart rhythm [30]. Hence, all subjects from the risk group were treated with acupuncture. Acupuncture treatment in our study included 12–15 sessions. It is important to mention that we chose a non-medicamentous way of treatment because, in the risk group, healthy subjects were included who were still showing no structural cardiac morbidity manifestations or arrhythmia, but in whose cardiorhythmograms pathological signs were identified. Consequently, there was no evidence to apply any pharmaceutical way of treatment.

Then followed stage C: in order to objectively appreciate the changes in the pathological signs under the influence of treatment, each person from the pathological group was subject to an HRV investigation immediately after the acupuncture course. Then this was compared with the initial investigation, the baseline measurement, done before the acupuncture treatment. Therefore, all investigations after the treatment course were analyzed by the standard linear and non-linear HRV analysis, and by the new physiological CRG method. The comparative analysis of HRV and the cardiorhythmogram of the measurements done before and after the treatment should answer the following questions: Do the non-steady-state events actually disappear after the treatment? How is the non-linear parameter of the CRG method, the counter-regulation, changing after the course? How does the HRV change after the course?

At the concluding stage, stage D, a comparative analysis between both methods of analysis, in order to generate physiological biomarkers, was made based on the identified pathological signs during cardiorhythmogram analysis, supported by the non-linear analysis.

## 3. Description of the Study Design

**Stage A**: Initial HRV measurement

I Standard linear HRV analysis:–Individuals with low HRV—the risk group;–Individuals with high HRV—the group of healthy;–NA individuals—those who could not be analyzed by these methods.

II Physiological cardiorhythmogram analysis:
–Individuals with identified pathological signs—the risk group;–Individuals without identified pathological signs—the group of healthy;

**Stage B**: To prove whether the pathological signs identified in stage A can be influenced clinically by a drug-free treatment

Acupuncture—a physiological drug-free treatment (included 12–15 sessions).

All subjects from the risk group were treated with acupuncture.

**Stage C**: Comparative measurement of HRV at the end of the acupuncture treatment course.

**Stage D**: Comparative analysis between standard linear HRV analyzing method and physiological method of cardiorhythmogram analysis in order to generate physiological biomarkers based on the identified pathological signs, supported by non-linear analysis.

Statistical analysis was performed by Fischer’s Exact Test. *p* < 0.05 was considered as significant, *p* < 0.01 means a very high significance. *p* > 0.05 was considered as not significant.

The present study was approved by the Research Ethics Committee at the session hold on 13 June 2016, the number of the document “Favorable Opinion of the Research Ethics Committee” is 78, dated 17 June 2016. The Chairperson of the Research Ethics Committee during the session was Prof. Viorel Nacu. The session took place at the “Nicolae Testemitanu” State University of Medicine and Pharmacy, Chisinau, Republic of Moldova.

## 4. Results

During the first stage, stage A, after the baseline measurement, all subjects (N = 218) were classified into the risk group and healthy group, based on two approaches to CRG analysis—the linear HRV method and the recent non-linear physiologic CRG method. In Figure 1, the comparative analysis of both methods which were used in this study in order to identify the pathological signs in cardiorhythmograms of healthy individuals is shown. Based on standard linear HRV analysis, 18% (N = 40) of individuals from the total amount N = 218 could be categorized into the risk group, whereas, 51% (N = 110) of individuals were categorized in the healthy group. In 68 individuals, the CRG could not be analyzed by the standard linear HRV method, so 32% (N = 68) of individuals were categorized into the non-available (NA) group. Based on the recent non-linear physiologic CRG method, 60% (N = 130) of individuals out of 218 were classified into the risk group, whereas 40% (N = 88) out of 218 were classified into the healthy group (Figure 1). By this approach, all CRGs could be analyzed, so that no one was classified into the non-available (NA) group.

Also, in the group of subjects who were classified by the linear methods initially as healthy, pathological signs were identified by the new physiological CRG method. Therefore, using the new method could also find those who were under the sensitivity threshold of the linear method. This is why the number of healthy individuals decreased from 110 to 88 (*p* < 0.01) after the physiological analysis. Correspondingly, the number of individuals with recognized risks increased from 40 to 133 (*p* < 0.01), as analyzed by the CRG method (Figure 1). In Figure 2 a schematical representation of the results obtained by applying the standard linear HRV method and the new physiological method of cardiorhythmogram analysis at every stage of the study: baseline measurement, after the treatment, and comparative analysis between both methods of analysis is shown (Figure 2).

Description of the scheme.

**Stage A**: baseline HRV measurement of 218 individuals

I Standard linear HRV analysis of 218
–Persons with low HRV—the risk group 40 (18%);–Persons with high HRV—the group defined as healthy 110 (51%);–NA Persons who could not be analyzed by this method 68 (32%).

II Physiological cardiorhythmogram analysis
–Persons with identified pathological signs—the risk group 130 (60%);–Persons without identified pathological signs—the group defined as healthy 88 (40%).

**Stage B**: To prove whether pathological signs identified in stage A can be influenced clinically by a drug-free treatment with acupuncture. All subjects from the risk group (the group with identified pathological signs) were treated by acupuncture, so 130 from 218 were treated. The treatment course included 12–15 sessions.

**Stage C**: Comparative measurement of HRV at the end of the acupuncture treatment course.

**Stage D**: Comparative analysis between standard linear HRV analyzing methods and non-linear, including the recent physiological method of cardiorhythmogram analysis in order to generate physiological biomarkers based on the ability of identification of the pathological signs during HRV and the cardiorhythmogram analysis.

## 5. Clinical Influence on the Detected Signs of a Pathological Heart Regulation

The comparative analysis of HRV and the cardiorhythmogram of the measurements done before and after the treatment showed the following dynamic of non-steady-state events: in 114 (*p* < 0.01) of the cases, the LF drops were not visible in the cardiorhythmograms after the treatment course. This means that, from 130 subjects with the detected pathological signs, treated by acupuncture, in 114 (88%) *p* < 0.01 subjects any pathological signs were observed after the treatment course. Of 130 treated subjects, 12% (16) of the subjects still continued to show pathological signs (Figure 3). Another important parameter, which was evaluated by the physiological CRG analysis before versus after treatment, is the counter-regulation: the frequencies of counter-regulation changed from the LF waves to the HF waves in 81% (105) *p* < 0.01 of the cases. Such a change in counter-regulation is regarded as one of the most important positive prognostic factors. From the remaining 12% (16) of the subjects who had no changes regarding the non-steady-state events’ dynamic of the LF drops and the counter-regulation, eight patients died. Representative cardiorhythmograms of two individuals, measured at baseline before and after the treatment (Figure 4). In Figure 4 the upper two CRGs belong to a patient in whose CRG the pathological signs disappeared after the treatment course and the other two CRGs (C, D) are from one of the 16 patients in who the pathological signs did not disappear after the treatment (deceased).

The physiological method of cardiorhythmogram analysis was objectively compared with standard linear and non-linear methods of HRV analysis. The non-linear method of HRV analysis more quantitatively described the changes in the cardiorhythmograms of patients before versus after treatment, in comparison to linear methods of HRV analysis. Among the linear methods which showed significant changes in the 114 patients, comparing cardiorhythmograms before and after treatment, were the following parameters (indicated as the mean values): meanNN changed from 890.15 to 947.43 (*p* < 0.05), sdNN changed from 40.23 to 47.82 (*p* < 0.01), sdaNN1changed from 15.16 to 24.03 (*p* < 0.01), rmssd changed from 32.28 to 38.69 (*p* < 0.01), and pNN50 changed from 0.13 to 0.19 (*p* < 0.01). In conclusion, the changes of the HRV by linear methods, comparing the measurement before and after the treatment course, were: the HRV increased significantly after the treatment in 75% (N = 97) *p* < 0.01 of cases. In 24% (N = 31) *p* < 0.05 of cases, the HRV remained without any changes. In 1.5% (N = 2) of cases the HRV decreased.

Among the non-linear methods which showed significant changes in the 114 patients, comparing cardiorhythmograms before and after treatment, were the following parameters (indicated as the mean values): Shannon changed from 1.95 to 2.15 (*p* < 0.001), FORBWORD from 13.96 to 19.48 (*p* < 0.01), FWSHANNON from 3.41 to 3.19 (*p* < 0.001), FWRENYI4 from 2.85 to 2.53 (*p* < 0.001), WSDVAR from 2.33 to 2.48 (*p* < 0.001), WPSUM02 from 0.02 to 0.01 (*p* < 0.01), and WPSUM13 from 0.41 to 0.51 (*p* < 0.001).

As can be seen from the comparative analysis of the cardiorhythmograms before and after the acupuncture course, it is evident that the pathological signs in the cardiorhythmograms found in this study can be detected by physiological CRG analysis. Secondly, pathological signs can be influenced clinically (Figure 4), without using drugs. Following the treatment with acupuncture, a large number (88%) N = 114 (*p* < 0.01) of subjects from the risk group could move to the healthy group after the treatment course. Like the results show (Figure 3), a progression of these pathological signs into structural heart pathology or arrhythmia could be prevented in the majority of cases.

## 6. Discussion

During the last 30 years, the morbidity and mortality of cardiovascular risks has drastically increased [1,23,32]. This fact has generated a challenging aim for researchers all over the world to find predictors for several cardiovascular risks. This scientific challenge can be solved interdisciplinarily, and the cooperation of physicians, mathematicians, physicists, and biostatistics is required. Since 2000, until now, a lot of attempts to propose predictors for several cardiovascular diseases have been made. Regarding the fact that, in every cardiovascular pathology, the regulatory cardiac system is involved, the approach of using non-linear dynamics in ECG and HRV analysis to find certain predictors has been used, e.g., it can be obviously observed, as shown by published studies on the prediction of atrial fibrillation and heart failure, by applying the non-linear method of analysis [15,17,23,25,26,32,33]. In these studies, there are different ECG features described as potential predictors. Zong et al. [25] showed, when observing a 30-min ECG, the feature which occurs before paroxysms of atrial fibrillation is the frequency of atrial premature contractions. In the study of Langley et al. [32], the possibility of predicting the recurrence of atrial fibrillation based on the assessment of a number of atrial and ventricular ectopic heartbeats is described. Therefore, they used 30-min RR interval data. A further prediction method regarding the appearance of recurrences of atrial fibrillation that is often described in the literature is the method of P-wave assessment on ECG [26,28,33]. In the studies which apply this method, researchers often take the duration of the P-wave into account, as well as its amplitude and the change in the P-wave, the intensities of the P-wave change power spectrum, and several non-linear P-wave measurements. In particular, Martinez A. et al. [33] describe the prediction of recurrence of atrial fibrillation, based on the P-waves, assessed using events observed one hour before the recurrence onset, as an effective tool. There are a lot of further examples. The findings of our study correspond to the literature, and the CRG features taken into consideration by the given study are also based on certain important pathophysiological mechanisms of heart regulation which stand at the background of the feature; these will be characterized further in this text. The advantages of the method proposed in this study consist in the possibility to recognize risks even in apparently healthy individuals, before the onset of disease or arrhythmia; therefore, they can be applied as predictors and, consequently, even measures of primary prophylaxis can be realized in the clinic. Another important advantage consists in the fact the CRG method can be assessed either by a trained physician, analyzing a CRG, or automatically, possessing a certain software, by evaluation of the CRG features via the non-linear parameter FWRENYI4.

The results of this study show that it is obvious that the physiological CRG method was more successful, in comparison with standard linear HRV analysis, in detecting pathological signs in cardiorhythmograms of apparently healthy individuals. Comparing the physiological method of cardiorhythmogram analysis with the standard linear HRV analysis, the recognition of pathological signs in the cardiorhythmograms of healthy individuals was about 60%, N = 130 from 218 cases (*p* < 0.01), compared to 18%, N = 40 from 218 cases (*p* < 0.05) recognized by the standard linear HRV analysis. This means that the sensitivity of the new method is three times higher in comparison with the existing method of standard linear HRV analysis. One of the reasons that explains such a remarkable difference is the limitation of the standard linear HRV analysis [16,18,19], which does not allow the standard HRV to analyze CRGs with events of non-stationarity, despite the fact that events of non-stationarity occurring in a steady-state CRG reveal predictively important information regarding the cardiac regulation [4,27,34,35]. The non-linear methods, including the new CRG method, do have the possibility to analyze such CRGs [15,25,27,28,29,32]. The limitations also belong to the factors, which explains why the number of the NA group was high in the case of linear HRV analysis. The NA group includes all of the ECGs which could not be analyzed by linear methods. This amounted to about 32%. This means that, in 68 out of 218 individuals, the CRGs could not be analyzed by using the linear HRV analysis. Taking into account that pNN50 can be, anyhow, assessed [13,16,18], it should be mentioned that, for a high number of the non-stationarity events, the LF drops and extra beats in the CRGs of the individuals of the NA group, a valid interpretation of pNN50 was also not possible [16,18,19]. In comparison, there was not one individual who could not be analyzed by using the physiological method, so that all 218 subjects of the study could be analyzed by using this method. To conclude, the new method, the physiological method of cardiorhythmogram analysis, allowed for the analysis even of those subjects who could not be assessed by the standard linear HRV method. The fact that, due to the CRG method, all cardiorhythmograms can be analyzed, shows the high reliability of the method. The number of individuals who were considered healthy by the linear method decreased after the analysis by the physiological method, from 110 to 88 individuals. This means that, by using the physiological method of CRG analysis, pathological signs can be recognized even in cases where the linear method does not detect these.

Validity is another very important characteristic of the recent CRG method, from a clinical point of view. It is manifested by the assessment results of the dynamics of pathological signs—the LF drops and HF counter-regulation before and after the treatment. Recently, it is one of the few methods which can assess the dynamic changes in the quality of waves’ frequencies in HF counter-regulation and LF drops [4,24,27]. It is clearly visible how precisely the new HF counter-regulation parameter evaluated the CRG before versus after the treatment. It could be recognized that the frequencies of counter-regulation changed from the LF waves to the HF waves in 81% (105) *p* < 0.01 of the cases. Such a change in the counter-regulation is regarded as one of the most important positive prognostic factors. Therefore, the physiological CRG method is a sensitive assessment tool which is able to deliver a valid evaluation of the changes in pathological signs even in healthy individuals. Furthermore, the pathological signs can serve as biomarkers for the test whether the risk disappeared or not, i.e., whether the treatment for this person was efficient or not—is the person out of the risk group or does he still belong to the risk group? LF drops is the second parameter of the CRG method. It makes risk stratification possible via its presence in a steady-state CRG. The assessment of its dynamic offers the possibility to understand whether the person is still in the risk group. The results of the present study show that, in 114 (*p* < 0.01) cases, the LF drops were absent in the cardiorhythmograms after the treatment course. This means that, after the treatment course for 130 subjects with the detected pathological signs, in 114 (88%) *p* < 0.01 individuals, any pathological signs were observed. Thus, the pathological signs can be influenced clinically by non–drug treatment. After the treatment, 93% of the individuals from the risk group were transferred to the healthy group. Therefore, the recognized cardiovascular risks in healthy individuals can be eliminated, preventing progression into structural heart disease or arrhythmia. Of 130 treated subjects, 12% (16) of the subjects still continued to show pathological signs (Figure 3 and Figure 4).

A very important proof of the validity of the new CRG method is the fact that, from the remaining 12% (16) of the subjects who had no changes regarding the non-stationary events’ dynamic of the LF drops and the counter-regulation, eight patients died. This also means that the pathological signs—the LF drops and HF counter-regulation—can be clinically applied as biomarkers for the identification of individuals who are at high risk. The important applicative value of the CRG method is represented on the one hand by risk prediction and on the other hand by the possibility of risk prevention. A further study with a higher number of individuals is needed to prove the predictive values of the pathological signs. The changes in the CRGs, which could be assessed by the recent CRG method, were also objectively proven by the non-linear parameter FWRENYI4. It validated the results of CRG method with high significance methodological differences between the standard linear method of HRV analysis and the physiological CRG method for the assessment of heart regulation.

For the linear method of analysis, a steady-state cardiorhythmogram (Figure 5) was analyzed by time-domain methods [13,16,18]. When in a cardiorhythmogram, non-steady-state events (Figure 6) occurred, such as the cardiorhythmogram having to be excluded from the analysis by the linear methods [16,18,19]. It is important to know that, especially in cardiorhythmograms, where non-steady-state events in rest state occur, important information is hidden regarding the detection of pathological signs in healthy individuals [4,10,15,19,22,23,24]. Such non-steady-state events in a rest-state cardiorhythmogram are described in the research work of N. Wessel [27]. In his work, these events are called “grosse Ereignisse” (dt.—big events). He found them in the resting state of patients who had an increased risk for developing a heart attack. In order to analyze them, he used non-linear methods of HRV analysis. In another study, the non-steady-state events in the cardiorhythmograms are called LF drops [4,24] and are described as risk factors for atrial fibrillation. In our study, these events in cardiorhythmograms were analyzed by the physiological CRG method [4] in order to find out whether these can be biomarkers for cardiovascular risks in healthy individuals.

The new physiological cardiorhythmogram analysis used here is based on the recently introduced LF drops and HF counter-regulation, which are non-linear, event-based dynamical HRV biomarkers [24]. The neural regulation of the heart rhythm is the physiological background of this method [4,5,6,14,20]. It is visualized by the corresponding waves’ structure in a cardiorhythmogram [4,24]. When analyzing a cardiorhythmogram by using physiological methods, there are some essential rules to be regarded. First, the cardiorhythmogram should be checked for whether it contains non-steady-state events (Figure 6), as the appearance of these in a rest-state cardiorhythmogram is characterized as a pathological sign [6,23,24,36]. From a pathophysiological point of view, this means that, in the rest state, the central neuronal heart regulation dominates the heart modulation versus the medullary regulation [4,9,10,20,37,38]. Using another way, the central neural heart regulation is increased to being pathologically high and the activity of the medullary heart regulation is insufficient [4,9,10,15,20,37,38]. Such a state should be recognized as early as possible because its progression can lead to structural heart diseases and arrhythmias [11,14,21,22,25,32].

In order to assess the level of progression of such a pathological state, the counter-regulation should be taken into account. The counter-regulation represents the part of the cardiorhythmogram following a non-stationarity event [4,24]. If the counter-regulation functions physiologically fine, it is driven by the parasympathetic part of the vegetative nervous system [4,20,21]. This is indicated by a certain wave structure (Figure 6, cardiorhythmogram A). In this case, in the cardiorhythmogram, high-frequency waves during the counter-regulation are recognized. Hence, the compensatory function of counter-regulation functions well. This means that it is able to compensate, during a certain period of time, the pathologically increased central neural heart regulation. The latter is presented by non-steady-state events in a cardiorhythmogram (Figure 6). If the counter-regulation was driven by the sympathetic part of the vegetative nervous system, this is considered as a pathological counter-regulation (Figure 6, cardiorhythmogram B) [4,21,34,35,37]. This means that the physiological compensation of the state of a pathologically increased central neural heart regulation cannot be compensated effectively enough [4,6,9,24,26,37] and is regarded as a risk factor for the appearance of arrhythmias, progression into structural heart diseases, and, in the worst-case scenario, death [6,11,14,22,23,24,25,27], as was also shown by the results of the present study.

In summary, the non-steady-state events identified in steady-state cardiorhythmograms of healthy subjects were regarded as pathological signs. These can be identified by physiological methods of HRV analysis. The pathological signs can be clinically influenced—after the treatment course, the pathological signs in the form of the non-steady-state events disappeared from the cardiorhythmogram, the HRV increased, and, in the majority of cases, the counter-regulation changed from sympathetically driven to parasympathetically driven. These changes led to the transfer of 88% (N = 114) of the subjects after the treatment from the pathological group to the healthy group. This has a very important clinical applicative value. Pathological signs can be identified in physiological cardiorhythmograms of healthy individuals in the early stages of the development of cardiologic pathology by applying non-complicated, easy, and quick methods of registration. Thus, they can be influenced by non-drug treatment methods in order to stop their progression into structural heart diseases or arrhythmia. Hence, this could be a real and easy application method for contributing to primary prophylaxis in cardiology.

The results show that it was possible to answer the question raised in this study in a positive way. Therefore, such pathological signs could further be defined as biomarkers for some cardiovascular risks and could, hence, serve for their prediction. This result’s applicative value would be important for the primary prophylaxis of arrhythmias, some structural heart diseases, and also a potential heart attack [8,11,12,22,23,24,27]. Moreover, the predictive effectiveness of such biomarkers could be tested in a separate study on a specific cardiological disease. FWRENYI4 is the non-linear parameter that makes a significant evaluation of the changes in the pathophysiological signs identified in CRG possible. Further studies should research whether it can be applied for supporting the automatic CRG assessment for risk prediction.

## 7. Limitations of the Study

The limitation of this study is that the new CRG method, based on evaluation of the heart’s regulation in the CRG and the recognition of signs for a pathologic heart regulation, depends on the skills of a trained physiologist. Although, in this study, the method was proven objectively and significantly by the non-linear parameter—the FWRENYI4, a further study to ensure the automaticity of the evaluation and prediction process is needed. The basis for such an automatized process is the non-linear parameter FWRENYI4. As was mentioned above, it approved, in this study, the evaluation technique by CRG method. Therefore, a further study should be done to test the predictive capacity of FWRENYI4 on a concrete clinical pathology.

## 8. Conclusions

In steady-state cardiorhythmograms of healthy individuals, pathological signs were identified that can be applied as biomarkers for the functional pathological state of the heart’s regulation.The efficacy of the new-found physiological method of cardiorhythmogram analysis for the recognition of cardiovascular risks in healthy individuals is significantly higher than the standard HRV analysis: using the new CRG method, the risk group detection is 60% versus 18% risk detection by the existing standard HRV linear analysis. Therefore, the sensitivity of the new method is three times higher in comparison with the existing method of standard linear HRV analysis.The pathological signs on which the new physiological approach to the cardiorhythmogram analysis is based can be defined as biomarkers for cardiovascular risk recognition in healthy individuals.The physiological CRG method is able to analyze even those cardiorhythmograms which show non-steady-state events.The pathological signs can be influenced clinically by non–drug treatment. After the treatment, 93% of the individuals from the risk group were transferred to the healthy group. Therefore, the recognized cardiovascular risks in healthy individuals can be eliminated, preventing progression into structural heart disease or arrhythmia.The non-linear parameter which significantly characterizes the dynamic of changes in the pathological signs in the CRGs before treatment versus after treatment is FWRENYI4.

## Figures and Tables

**Figure 1 entropy-25-01023-f001:**
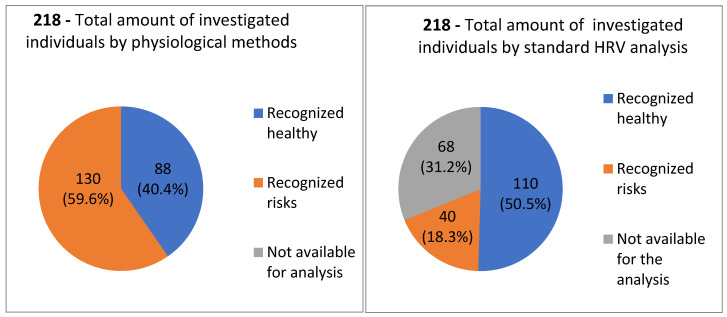
Results of baseline cardiorhythmograms analysis by the physiological method (**left**) and by the standard linear HRV method (**right**). Based on this analysis, the individuals were divided in two groups: the healthy group and the group with recognized risks.

**Figure 2 entropy-25-01023-f002:**
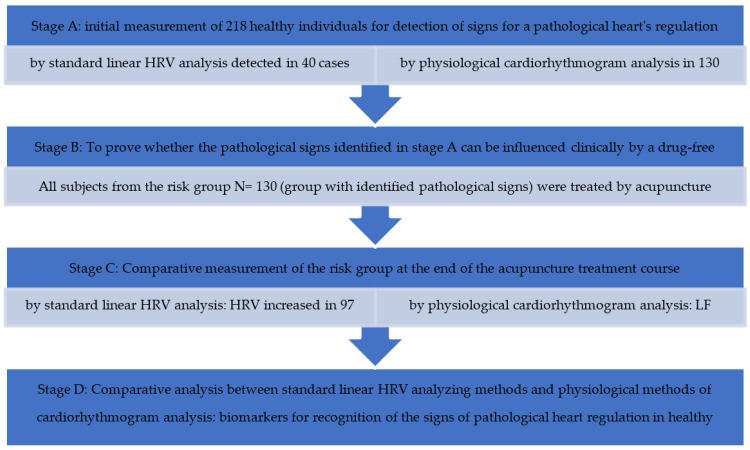
Scheme of results obtained by applying the standard linear HRV method and the new physiological method of cardiorhythmogram analysis at every stage of the study: baseline measurement, after the treatment, and comparative analysis between both methods of analysis.

**Figure 3 entropy-25-01023-f003:**
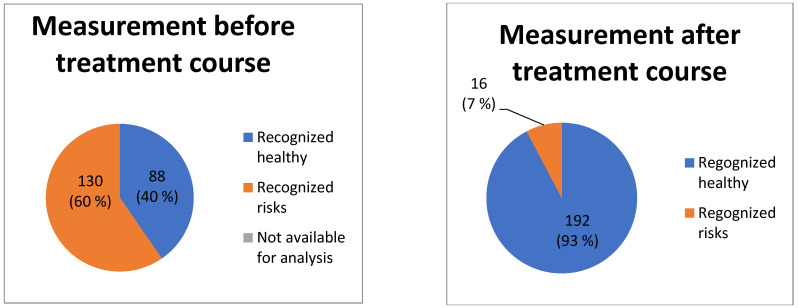
Results of comparative analysis between the CRGs before and after the treatment course, evaluating the presence of pathological signs. The effectiveness of the treatment course is clearly visible. The number of patients belonging to the risk group is reduced by eight times—from 130 individuals divided into the risk group before the treatment, the number decreased after the treatment to 16 (*p* < 0.01) individuals remaining in the risk group.

**Figure 4 entropy-25-01023-f004:**
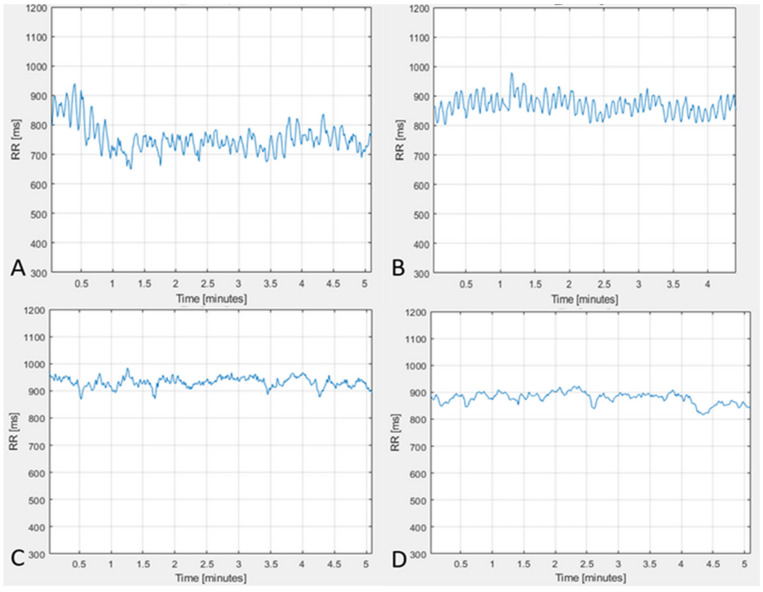
Representative cardiorhythmograms of two individuals, measured at baseline before and after the treatment. The CRGs (**A**,**B**) belong to a patient in whose CRG the pathological signs disappeared after the treatment course. The CRGs (**C**,**D**) are from one of the 16 patients in who the pathological signs did not disappear after the treatment (deceased). Both (**A**,**C**) show pathological signs in the form of LF drops, followed by a pathological counter-regulation. After the treatment course in (**B**), the pathological signs disappeared. In (**D**), the pathological signs still remain.

**Figure 5 entropy-25-01023-f005:**
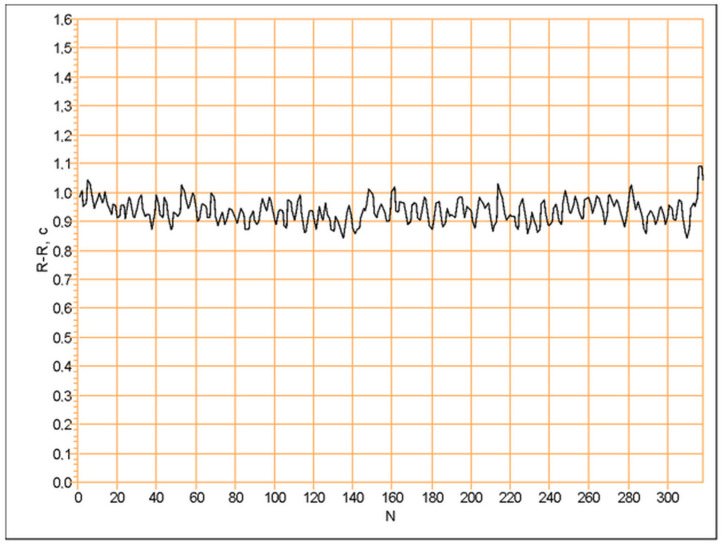
Steady-state cardiorhythmogram without events of non-stationarity, suitable for standard linear HRV analysis.

**Figure 6 entropy-25-01023-f006:**
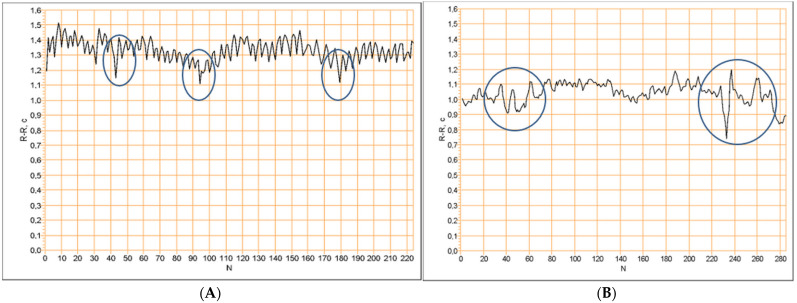
Steady-state cardiorhythmograms where events of non-stationarity are presented. Cardiorhythmogram (**A**): the pathological signs are presented by the LF drops (encircled), but they are followed by a physiological counterbalancing via HF waves, so a low risk is estimated. Cardiorhythmogram (**B**): LF drops (encircled) are present followed by a pathological counter-regulation, predominant by LF waves, high risk is estimated.

## Data Availability

Original data are available upon request from the first author.

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
