# Peer review of "Pathological Heart Rate Regulation in Apparently Healthy Individuals"

_entropy, 2023, doi:10.3390/e25071023_

Round 1
Reviewer 1 Report
It is a very nice paper. It has nice flow to it which makes it easy to read.
Major comments:
1. If I understood clearly, this paper does not present some new non-linear HRV measures but uses some existing. This could be made more explicit in the paper.
2. I do not see how it follows that in the phase A your HRV analysis detects pathological signs. Since patients have been treated with acupressure, that just means you detected some properties in the HRV that can be influenced by acupressure. This, it seems to me, needs to be clarified in the paper.
Minor comments:
1. Abstract is too long for my taste.
2. Figures 1 and 4 could be made more compact and self-contained.
3. The other figures could be more compact.
Author Response
Thank you very much for your report and your valuable suggestions. We now explicitly state that we have used existing non-linear HRV parameters. In addition, we now state that we were able to use CRG parameters to identify certain features in HRV that can be influenced by acupuncture.
Minor:
We have shortened the summary and removed Fig. 1 as it was redundant. We have also corrected the appearance of the other figures.
Reviewer 2 Report
The idea of the study is perfect. Still, it is difficult to follow the idea due to huge text paragraphs. For example, paragraphs from lone 48 to 87, 123-170, 337-403, 431-461. I'd strongly recommend the authors to subdivide these paragraphs into smaller ones. In a whole, I would support this study, as it focuses the audience on important aspect of heart regulation and, in general, on the notion of "normality" in heart regulation.
Explanation of FORBWORD, FWSHANNON, FWRENYI4, WSDVAR, WPSUM02, and WPSUM13 parameters is needed as not all readers are familiar with them. Referencing some studies relating to these parameters is not enough. Especially "FWRENYI4" must be discussed more profoundly as it preseneted potential diagnostic value.
For example, "Based on the recent non-linear physiologic CRG method, 60 % (N = 130) individuals out of 218 were divided into the risk group, whereas 40 % (N = 88) out of 218 were divided into the healthy group (fig. 3)".
I don't like "were divided'. Better "were allocated" or "were classified as", "were considered as", and suchlike.
Author Response
Thank you very much for your very helpful report. We introduced subsections for the huge text passages from line 48 to 87, 123-170, 337-403, 431-461. We have included now the explanation of FORBWORD, FWSHANNON, FWRENYI4, WSDVAR, WPSUM02 and WPSUM13.
The cited sentence was changed to
"Based on the recent non-linear physiologic CRG method, 60 % (N = 130) individuals out of 218 were classified into the risk group, whereas 40 % (N = 88) out of 218 were divided into the healthy group (fig. 3)".